# HDFlow: Enhancing LLM Complex Problem-Solving with Hybrid Thinking and Dynamic Workflows

## Abstract

Despite recent advancements in large language models (LLMs), their performance on complex reasoning problems requiring multi-step thinking and combining various skills is still limited. To address this, we propose a novel framework HDFlow for complex reasoning with LLMs that combines fast and slow thinking modes in an adaptive manner. Our approach consists of two key components: 1) a new approach for slow, deliberate reasoning called Dynamic Workflow, which automatically decomposes complex problems into more manageable sub-tasks and dynamically designs a workflow to assemble specialized LLM or symbolic reasoning tools to solve sub-tasks; 2) Hybrid Thinking, a general framework that dynamically combines fast and slow thinking based on problem complexity. Finally, we propose an easy-to-scale method for automatically synthesizing a large-scale dataset of 27K challenging reasoning problems for complex reasoning and a hybrid thinking tuning method that trains smaller LLMs on this dataset to internalize the fast/slow hybrid reasoning strategies. Experiments on four reasoning benchmark datasets demonstrate that our slow thinking with dynamic workflows significantly outperforms Chain-of-Thought, and hybrid thinking achieves the highest accuracy while providing an effective balance between computational efficiency and performance. Fine-tuning using our hybrid thinking approach also significantly boosts the complex reasoning capabilities of open-source language models. The results showcase the promise of slow thinking, dynamic workflows, and hybrid thinking in expanding the frontier of complex problem-solving with LLMs[1].

## 1 Introduction

Large language models (LLMs) have demonstrated remarkable capabilities across a wide range of tasks, from code generation and mathematical reasoning to natural language understanding and generation. Recent advancements in symbolic reasoning and tool usage, such as AlphaGeometry (Trinh et al., 2024) and AlphaProof (AlphaProof/AlphaGeometry teams), have shown significant improvements in specific domains by integrating LLMs with specialized procedures and symbolic reasoning engines. Various prompting strategies, such as Chain-of-Thought (CoT) (Wei et al., 2022), Tree of Thoughts (ToT) (Yao et al., 2024), and Graph of Thoughts (GoT) (Besta et al., 2024a), have been developed to enable different reasoning topologies to enhance LLM problem-solving capabilities. However, enhancing the reasoning abilities of LLMs to solve complex problems across various domains in a unified framework remains a challenge for expanding their real-world applicability.

First, complex problem-solving often requires combining various knowledge domains, skills, and tool usage. While previous approaches such as AlphaCodium (Ridnik et al., 2024) and Alphageometry (Trinh et al., 2024) have demonstrated the potential of combining language models and symbolic reasoning to solve complex problems, they rely on manually designed workflows tailored to specific domains (i.e., competitive programming or geometry theorem proving). The language model and symbolic engine take predefined turns in a rigid problem-solving process. This limits the applicability and adaptability of these systems to broader domains. Thus, we aim to enhance the generic problem-solving capabilities of LLMs by dynamically alternating between natural language

---

[1]Code and data will be released on Github.

reasoning in the "text space" and symbolic reasoning in the "symbolic space" based on the problem at hand. This dynamic integration of the two reasoning modes enables the system to address a much broader range of problems and adapt the problem-solving process to the unique requirements of each task. Second, traditional approaches to complex problem-solving with LLMs often rely on a single mode of thinking, which may struggle with more intricate tasks that demand a deliberate, analytical approach. For example, many approaches employ a fixed reasoning strategy, such as CoT prompting, regardless of the problem's complexity. For instance, OpenAI's most recent o1 model[2] only engages in a singular deep thinking mode despite the complexity of the user's query. This can lead to suboptimal performance on tasks requiring varying reasoning levels. The model may either over-commit resources on simple tasks or underperform on more complex ones. While multi-agent frameworks such as AutoGPT (Significant Gravitas), ReAct Yao et al. (2022), and AutoGen (Wu et al., 2023) have addressed some aspects of this challenge by enabling recursive goal decomposition, interleaving reasoning and acting, and state-driven workflows, they do not fully exploit the potential of thinking approaches that can switch between intuitive thinking and more analytical thinking modes based on problem complexity. Finally, as problem complexity increases, the performance of existing approaches tends to degrade significantly, highlighting the need for frameworks that can scale to handle even the most challenging reasoning problems. Recently, OpenAI o1 model (OpenAI, 2024) demonstrates the potential to consistently improve LLM performance of complex reasoning with compute scaling in inference-time through deep thinking.

To address these limitations, we propose a novel framework for complex reasoning with LLMs that combines fast (system I) and more analytical slow thinking (system II) adaptively, inspired by the dual process theory of human cognition (Daniel, 2017). Our approach consists of two key components. First, we introduce a new approach for slow, deliberate reasoning called **Dynamic Workflow**, which automatically decomposes complex problems into more manageable sub-tasks. It then dynamically designs a workflow to assemble specialized LLM or symbolic tools to solve each sub-task. To achieve this, the dynamic workflow orchestrates a team of specialized LLM experts, each contributing unique domain knowledge or tool usage, to solve the sub-tasks in a structured manner. Second, we propose **Hybrid Thinking**, a general framework that dynamically combines fast and slow thinking based on problem complexity. For simpler tasks, the model defaults to a fast-thinking mode using CoT strategy. When the model's confidence in the fast thinking output is low, it automatically switches to slow thinking with dynamic workflow, allowing for more efficient and more accurate problem-solving. Finally, to train local LLMs for complex reasoning, we present an easy-to-scale method for automatically synthesizing a large-scale dataset of 27K challenging reasoning problems and propose a hybrid thinking tuning approach that finetunes open-source LLMs on this dataset, enabling them to internalize the fast/slow hybrid reasoning strategies.

We conduct experiments on four reasoning benchmark datasets (i.e., BBH (Suzgun et al., 2022), MATH (Hendrycks et al., 2021), Game of 24 Yao et al. (2024), DeepMind Math (Saxton et al., 2019). Experiments using GPT-4-Turbo reveal that slow thinking with dynamic workflows significantly outperformed CoT, with an average accuracy improvement of 22.4%. Hybrid thinking, which combines fast and slow thinking, achieved the highest accuracy on three of the four datasets and struck an effective balance between computational efficiency and performance. Furthermore, fine-tuning Llama-3-8B-Instruct using hybrid thinking significantly boosted performance across all datasets compared to the original model. Overall, the results demonstrate the promise of slow thinking with dynamic workflows and hybrid thinking in enhancing the complex problem-solving abilities of LLMs.

## 2  RELATED WORK

**Symbolic Reasoning and Tool Usage.** Bridging LLMs with symbolic reasoning and tool usage has demonstrated significant improvements across various domains. AlphaCode (Li et al., 2022; AlphaCode Team) combines LLMs with a specialized search and reranking mechanism, achieving top-tier performance in competitive programming. AlphaCodium (Ridnik et al., 2024) improves AlphaCode's performance by applying a predefined multi-stage process of problem analysis, solution generation, and iterative testing and bug fixing. By using an evolutionary search procedure guided

---

[2]o1-preview model tested on Sept.24, 2024. o1-preview model thinks for a few seconds to users' casual conversational queries such as *How are you?*

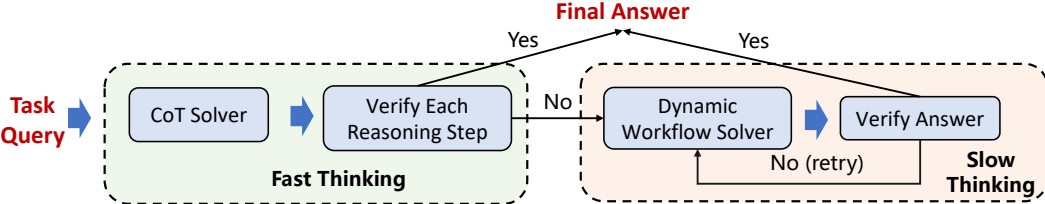

Figure 1: Overview of our HDFlow approach for complex problem-solving. Overall, it is a dual-path hybrid thinking approach, beginning with a CoT solver for initial fast reasoning followed by verification of each reasoning step. If verification fails, the process transitions to a slower, more deliberate "Dynamic Workflow Solver." This solver iterates until a verified answer is obtained, incorporating a final verification step before concluding with a solution.

by an LLM, FunSearch (Romera-Paredes et al., 2024) can discover new mathematical constructions and algorithmic heuristics. AlphaGeometry (Trinh et al., 2024) leverages a neuro-symbolic system trained on synthetic data to guide a symbolic deduction engine, achieving near-expert performance in geometry theorem proving. Program of Thoughts (PoT) (Chen et al., 2022) disentangles computation from reasoning by expressing the reasoning process as a program. Chain of Code (Li et al., 2024) encourages LLMs to write pseudocode for challenging sub-problems, which is then executed by the LM itself when a standard interpreter cannot handle it.

**Prompting Strategies.** Various prompting strategies have been developed to enable different reasoning topologies (Besta et al., 2024b) for enhancing LLM problem-solving capabilities. Chain-of-Thought (CoT) prompting (Wei et al., 2022) first introduced the concept of generating intermediate reasoning steps to improve performance on complex tasks. Building upon this, the Tree of Thoughts (ToT) (Yao et al., 2024) enables the exploration of multiple potential reasoning paths and incorporates deliberate decision-making through self-evaluation and backtracking. Graph of Thoughts (GoT) (Besta et al., 2024a), models LLM-generated information as an arbitrary graph where thoughts are vertices and dependencies are edges. SELF-DISCOVER (Zhou et al., 2024) introduces a self-discovery process where LLMs select and compose multiple atomic reasoning modules into explicit reasoning structures.

**Multi-Agent Frameworks for Task-Solving.** Recent advancements also have led to the development of multi-agent collaboration for complex tasks. AutoGPT (Significant Gravitas) pioneers recursive goal decomposition and sequential task completion using LLMs. ReAct (Yao et al., 2022) introduces interleaving reasoning and acting, allowing LLMs to generate both reasoning traces and actions. Reflexion (Shinn et al., 2024) enhances these agents with verbal reinforcement learning for improved decision-making. MetaGPT (Hong et al., 2024) incorporates human workflows and SOPs to enable domain-specific multi-agent collaboration in software tasks. AutoGen (Wu et al., 2023) creates a flexible framework for customizable agent conversations with human input, while CAMEL (Li et al., 2023) uses role-playing to foster autonomous cooperation. StateFlow (Wu et al., 2024) conceptualizes task-solving as state-driven workflows for better control. In contrast, our approach uniquely integrates hybrid thinking, combining fast and slow modes with automated workflows to enhance LLMs' adaptability and complex reasoning.

## 3 OVERVIEW OF THE HYBRID THINKING APPROACH

Figure 1 illustrates our hybrid thinking approach. It combines the strengths of fast and slow thinking modes to enable LLMs to more effectively solve complex reasoning problems, which consists of the three key components. 1) **Fast Thinking with Direct CoT.** In the fast thinking mode, the LLM uses a direct chain of thought (CoT) approach to quickly solve the task query if possible. This leverages the LLM's core abilities to perform certain types of reasoning efficiently by directly generating the rationale and the final answer. 2) **Adaptive Combination of Fast and Slow Thinking.** Next, we employ a self-verification mechanism where the LLM examines each step of the fast-thinking CoT reasoning to assess its confidence in the generated answer. This is achieved by applying the LLM to analyze the coherence, logical consistency, and correctness of each reasoning step. If the LLM detects any inconsistencies, errors, or low-confidence steps during this self-verification process, it

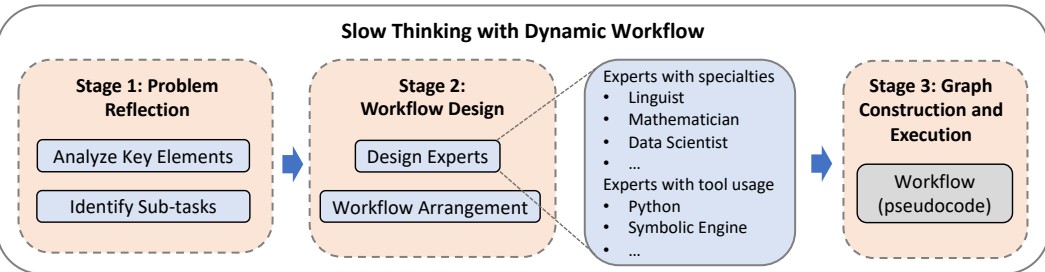

Figure 2: Three-stage framework of dynamic workflow. The dynamic workflow design begins with Problem Reflection, where key elements are analyzed and sub-tasks identified. Stage 2 focuses on Expert Design, utilizing a variety of specialists and tools to architect an optimal workflow. Stage 3 involves constructing and executing the workflow graph to get the final result.

triggers a switch to the slow-thinking mode. 3) **Slow Thinking with Dynamic Workflow.** To tackle highly complex tasks, we propose a novel slow-thinking mechanism called Dynamic Workflow (Figure 2), which automatically decomposes the original task into sub-tasks and dynamically switches between verbal reasoning and symbolic reasoning to solve each sub-task. Our approach starts with multi-level problem reflection and decomposition. It then designs a workflow to assemble specialized LLM skills or symbolic tools for sub-tasks. Next, we dynamically chain together the sub-task reasoning steps into a multi-step workflow and execute the workflow. Finally, all sub-task results are aggregated into the final answer to the original query. We will present details in Section 4.

By first attempting fast thinking, our hybrid thinking approach can efficiently handle queries that are within the LLM's core capabilities. When the query exceeds what fast thinking alone can confidently handle, the hybrid thinking will smoothly transition to a slow thinking workflow to tackle a broader range of challenges accurately.

# 4 SLOW THINKING WITH DYNAMIC WORKFLOW

In contrast to the rapid responses of fast thinking (e.g., CoT), our new slow-thinking mechanism applies dynamic workflow to enable a more deliberate, analytical approach to complex problem-solving (see Figure 2). It allows an LLM to dynamically transition between reasoning in the text space (natural language reasoning) and the symbolic space (symbolic reasoning). The high-level idea is we first let the LLM decompose the original reasoning problem into several more manageable sub-tasks and solve each sub-task to form the final solution. When necessary, the LLM Engine will translate the sub-problem from the text space into the symbolic space, enabling the symbolic engine[3] to perform precise symbolic reasoning. The results are then mapped back into natural language using the LLM Engine. By decomposing the problem, combining the strengths of both natural language and symbolic reasoning in a tailored workflow, and executing it from start to finish, LLMs can tackle very hard problems that require multiple steps of accurate reasoning. Appendix B presents a complete example solution using our dynamic workflow approach and compares with the solution using OpenAI o1-preview. Prompts used are listed in Appendix C.

## 4.1 BREAKING DOWN COMPLEXITY: PROBLEM ANALYSIS AND DECOMPOSITION (STAGE 1)

The first step in our slow thinking is problem analysis and planning. We aim to break down the original problem statement into more manageable sub-tasks. Specifically, the LLM is asked to analyze the key elements of the query, such as available information, constraints, and the desired output. It then identifies logical sub-goals needed to progress from the initial state to the solution. This decomposition allows the LLM to approach the problem in a structured manner, focusing on one part at a time. Therefore, the LLM can catch gaps in reasoning and handle complex problems that the fast thinking of CoT alone would struggle with.

---

[3]In this paper, we mainly use program to achieve symbolic reasoning.

**Problem Reflection.** The first step in tackling complex problems is conducting a thorough problem reflection. This involves the LLM analyzing the original problem and restating it in its own words to demonstrate understanding. Our problem reflection includes two parts: 1) Identifying the core objective or question posed by the problem. 2) Recognizing any constraints, assumptions, or special conditions mentioned. By internalizing the problem through reflection, the LLM can gain a solid understanding of what needs to be accomplished before proceeding to decomposition.

**Subtask Decomposition.** Once the problem is well understood, the LLM is instructed to perform a multi-level decomposition to break it down into some tractable sub-problems. The LLM is asked to follow four principles to achieve an optimal decomposition. *Sequential dependency.* The sub-problems are organized in a logical sequence, such that the outputs of earlier steps feed into subsequent ones, creating a structured workflow from start to finish. *Non-overlapping.* Each sub-problem represents a distinct portion of the original problem, with no duplication of work between sub-problems. This keeps the overall solution efficient. *Proper Decomposition.* The sub-problems are decomposed to the optimal level of granularity - not so small that there are too many to track and coordinate, but not so large that they are still struggling to solve. *Modular.* Where appropriate, sub-problems are defined in a generalizable, modular way, such that the logic and code used to solve them can potentially be reused to solve similar problems in other contexts.

**Integrating Symbolic Reasoning.** Another key aspect of our approach is leveraging the symbolic engines to modularize the solution and handle well-defined sub-tasks more accurately. For example, some sub-tasks in the decomposition can often be addressed by writing code functions. Therefore, we explicitly instruct the LLM to consider sub-tasks that can be well handled by writing and executing modular code in subtask decomposition.

## 4.2 ORCHESTRATING EXPERTISE: WORKFLOW DESIGN (STAGE 2)

With the problem decomposed into sub-tasks, our approach next proposes a team of specialized experts, each contributing unique skills and tools, arranged in a dynamic workflow. The central component is a Meta-Expert, initialized from the foundation LLM, designs the expert team, and coordinates their efforts. The orchestration process consists of four steps.

1. **Design of Experts.** Based on the identified sub-tasks, the Meta-Expert designs a team of specialized experts with one expert solving one sub-task. Each expert is assigned a unique name and a clear description of their specific skills, knowledge, and responsibilities[4]. The dynamic workflow leverages two types of experts to handle each sub-task, enabling a seamless integration of verbal and symbolic reasoning. The first type are specialized experts initiated from LLMs, such as linguists, mathematicians, and data scientists. These experts bring domain-specific knowledge and skills to the workflow, allowing for sophisticated verbal reasoning and analysis within their fields. The second type of expert focuses on symbolic reasoning, particularly using programming or other symbolic engines[5]. For example, some sub-tasks can often be addressed by writing compact, targeted code functions. This allows the LLM to handle common operations such as mathematical calculations, data parsing and manipulation, and so on without bringing errors.

2. **Workflow Arrangement.** The Meta-Expert arranges the experts into an efficient workflow sequence. Each expert's output serves as the input for the next, progressively moving towards the final solution. The Meta-Expert ensures there is no redundancy of functions across experts.

3. **Collaboration and Iteration.** As the experts work through the problem, the Meta-Expert facilitates collaboration and puts together their inputs and outputs. For sub-tasks involving logical reasoning, mathematical operations, data structures, or programming, the Meta-Expert provides strategic guidance and sends the implementation details to the corresponding symbolic reasoning experts. These experts utilize LLMs to generate code, which is then executed to perform symbolic reasoning in Stage 3.

---

[4]Our implementation leverages JSON for efficient data management and extraction across the system.

[5]We mainly use Python code interpreter as the symbolic engine in our experiments, but our approach can be extended to other symbolic engines, such as the symbolic deduction engines used in AlphaGeometry (Trinh et al., 2024) to solve Euclidean geometry problems.

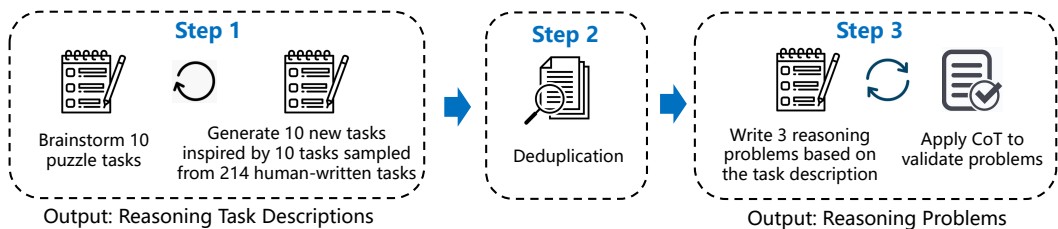

Figure 3: Data synthesis of complex reasoning problems.

4. **Final Review and Conclusion.** The last expert in the workflow, often an LLM specialist, is tasked with holistically reviewing the findings of the previous experts and generating the final answer to the original problem.

By combining the power of specialized LLMs and the usage of tools into a thoughtfully designed, adaptable workflow, our approach can tackle complex problems that are beyond the capabilities of the original model.

## 4.3 Flow Execution: Constructing and Running Workflows (Stage 3)

With the workflow graph generated, our approach finally proceeds to execute the graph to get the final result. The execution follows the dependency order, ensuring the correct flow of data between experts. To ensure robust execution, if any of the generated code encounters errors, the corresponding symbolic reasoning experts will trace the issue, use the error message to repair the code, and rerun it. As the workflow progresses, the downstream experts continually update their memory with the intermediate results and insights generated by previous experts. Upon completion of the workflow execution, the last LLM expert analyzes the results, identifies key findings, and summarizes them into a final answer to the original problem. The workflow execution is not a one-time process. The LLM continually assesses the quality and correctness of the final generated solutions and identifies potential errors. It engages in iterative rerun by applying a different problem decomposition, expert assignments, or adjusting the workflow structure.

## 5 Model Tuning of Hybrid Thinking

In our experiments, we observed that open-source language models (typically those with around 7B parameters) often struggle with advanced meta-planning and problem-solving skills required for solving difficult reasoning tasks. To address this limitation and develop local smaller models with hybrid thinking abilities comparable to the large models, we construct a comprehensive training dataset and propose hybrid thinking tuning to improve the complex reasoning abilities of local models. We define "local" models as models that can be trained and deployed on local hardware with limited computational resources, such as the Llama-3 model (Meta, 2024). The primary challenge lies in constructing a large-scale dataset of reasoning problems that are sufficiently diverse, high-quality, and difficult. Such a dataset is crucial for teaching smaller local models to perform complex reasoning tasks. However, manually curating such a dataset presents significant difficulties in ensuring a wide range of problem domains and maintaining high standards in problem formulation. As a result, it is extremely time-consuming and expensive to ask human experts to consistently generate problems meeting all criteria. Therefore, we propose a novel approach for automatically generate a variety of reasoning problems and collect solutions of hybrid thinking, which can then be used to train our local LLMs.

### 5.1 Reasoning Problems Synthesis

To enhance reasoning task diversity and coverage, our data synthesis pipeline consists of three steps (Figure 3). In the first step, we strategically leverage human-authored seed tasks to inspire the creation of new reasoning problems (similar to Self-Instruct (Wang et al., 2023)) or let the LLM brainstorm reasoning puzzles that cover a variety of task formats, difficulty levels, and problem domains. This step only focuses on generating high-level task descriptions to encourage diversity.

**Interpret a Morse Code Message**: Given a string of Morse code, translate it into English text, adhering to standard Morse code conventions. The task involves recognizing each sequence of dots (.) and dashes (-) as letters and spaces as separators for words.
A Morse code sequence has been found etched into an old artifact. It is believed to be a significant mathematical formula. The Morse code is: '-. .. -. . - -.– / - .... .-. . . / - .. − . ... / ... . ...- . -. - -.– / ..-. .. ...- . / . .-.- ..- .- .-.. ... / — -. . / .... ..- -. -.. .-. . -.. / .-.- -.. / .-.- .-.. / - .- .- . -. - -.– / - .... .-. . .'.
Decode this Morse code into English text, adhering to the standard Morse code conventions where sequences of dots (.) and dashes (-) represent letters, and spaces are used to separate words.

---

**Cryptarithm Task: Solve the Equation**: In this cryptarithm, each letter represents a unique digit from 0-9: **CROSS + ROADS = DANGER** No number may begin with zero. Determine the digit each letter represents to satisfy the equation.

---

In a game of spies, two teams use different substitution ciphers to communicate. Team A uses a cipher where each letter is replaced by the letter three positions to the right in the alphabet (with wrapping), while Team B uses a cipher where each letter is replaced by the letter four positions to the left (with wrapping). During the game, a message encrypted using Team B's cipher was intercepted: "XLMW MW XLI GIRXVI." Decode this message assuming it was meant for Team A but encrypted by Team B.

Figure 4: Three example reasoning problems generated by our data synthesis approach.

In the second step, we apply deduplication to remove near-identical tasks. Finally, we apply LLMs again to write three specific problems based on the task descriptions and validate those problems.

**Task Generation Inspired by Seed Tasks.** The first step of our reasoning data synthesis pipeline is generating an expanded set of reasoning tasks. We augment the few-shot prompts with 10 high-level task descriptions randomly sampled from the 214 BigBench tasks (Srivastava et al., 2022). Next, we employ the 10 seed tasks as in-context examples to prompt LLMs[6] to generate 10 new tasks. To encourage additional diversity in the generated tasks, we also let the LLM to brainstorm different genres of puzzles, such as crossword puzzles, math puzzles, number puzzles, relational puzzles, logic puzzles, etc. By repeating two strategies, we produce an expanded pool of 45K candidate reasoning tasks that creatively cover diverse reasoning types and scenarios.

**Data Filtering and Deduplication.** The previous task generation step produces a sizable pool of candidate reasoning tasks. However, the generated data is likely to contain duplicate or highly similar entries. To address this, we employ a comprehensive data filtering and deduplication process. First, we apply n-gram to identify nearly identical tasks. Next, we filter out any tasks or problems that fail to meet our quality criteria by prompting GPT-4-Turbo, such as insufficient complexity (e.g., trivial one-step questions), or ambiguity in the description. This helps ensure that only high-quality, unambiguous reasoning tasks are retained in the final dataset. Through this rigorous deduplication and filtering process, we condense the pool of 45K generated tasks down to 18K deduplicated tasks.

**Reasoning Problem Synthesis.** In the last step, we aim to synthesize multiple concrete reasoning problems for each of the 18K tasks. Taking each task's description as input, we prompt an LLM to generate 3 distinct questions or problems that test the specified reasoning skill. This enables us to turn each high-level task into a set of actual solvable questions, resulting in a pool of 54k reasoning problems. To ensure the generated problems are well-posed and solvable, we employ a chain-of-thought (CoT) based validation step. We prompt GPT-4-Turbo to apply CoT to each synthesized problem and analyze if the resulting reasoning steps coherently lead to a definite answer. Problems for which the model fails to converge to a clear solution or exhibits inconsistent reasoning are filtered out. This results in the final 27K reasoning problems. Figure 4 provides three examples of reasoning problems generated.

## 5.2 FINETUNING OPEN-SOURCE MODELS ON SYNTHESIZED DATA

To prepare the training data for enhancing the open-source models' complex problem-solving abilities, we utilize the GPT-4-turbo model to collect reasoning trajectories on the dataset of synthesized

---

[6]We use both GPT-4-0125 and Claude-3-Opus to encourage diversity. We find Claude-3-Opus does generate very different reasoning tasks compared with GPT-4-0125.

Table 1: Accuracy (%) of GPT-4-Turbo-0125 across different reasoning modes on various datasets. We show the accuracy of the model using Chain of Thought (CoT) v.s. slow thinking (with dynamic workflow) and Hybrid Thinking approaches proposed by us. The Fast/Slow indicates the ratio of Fast and Slow Thinking contributions in the Hybrid approach. Results are derived from the top 100 instances for each sub-category in BBH (27 sub-tasks), MATH (7 sub-domains), and GameOf24 (3 difficulty levels) to reduce API cost and ensure replicability. For the DeepMind Math dataset, the top 10 instances from each of the 56 sub-domains were used.

| Methods | BBH | MATH | DeepMind Math | GameOf24 | Avg. |
|---|---|---|---|---|---|
| CoT (Fast Think.) | 77.8 | 62.6 | 53.4 | 9.3 | 50.8 |
| Slow Think. | 87.1 (+9.3) | 67.6 (+4.6) | **67.7** (+14.3) | 70.3 (+61.0) | 73.2 (+22.4) |
| Hybrid Think. | **87.8** (+10.0) | **70.0** (+7.9) | 59.6 (+6.2) | **72.0** (+62.7) | 72.4 (+21.6) |

Table 2: Average number of inference tokens of GPT-4-Turbo-0125 using different reasoning modes on various datasets. Performance is reported in Table 1.

| Methods | BBH | MATH | DeepMind Math | GameOf24 | Avg. Tokens |
|---|---|---|---|---|---|
| CoT (Fast Think.) | 351 | 992 | 581 | 387 | 577.8 |
| Slow Think. | 3227 | 5694 | 3562 | 5246 | 4432.0 |
| Hybrid Think. | 1299 | 4398 | 1742 | 4983 | 3105.5 |

and mathematical problems. For each problem, GPT-4-turbo generates one or several fast/slow reasoning trajectories using the hybrid thinking approach. Each reasoning trajectory consists of a sequence of (query, answer) pairs representing the model's step-wise hybrid thinking process. Therefore, we use all (query, answer) pairs from the reasoning trajectories to construct the training data, capturing the complete problem-solving process. When multiple reasoning trajectories are produced (iterative retry), only the solution trajectory that passes the verification process is retained in the training set to optimize the model's problem-solving capabilities, while the verification results for all trajectories are kept to enhance the model's self-verification abilities. We choose the Llama-3-8B-Instruct model (Meta, 2024) as the foundation model for our hybrid thinking tuning experiments. More training details are included in Appendix A.

## 6 EXPERIMENT

### 6.1 REASONING BENCHMARK DATASETS

**BIG-Bench Hard (BBH)** (Suzgun et al., 2022): A subset of 27 challenging tasks from the BIG-Bench benchmark (Srivastava et al., 2022), which aims to measure the capabilities and limitations of language models. **MATH** (Hendrycks et al., 2021): A dataset consisting of 5,000 test problems from mathematics competitions across seven disciplines. **Game of 24** (Yao et al., 2024): A mathematical reasoning challenge dataset containing 1,362 games sorted by human solving time. The goal is to use four given numbers and basic arithmetic operations (+ - * /) to obtain 24. **DeepMind Math** (Saxton et al., 2019): A dataset consisting of various types of mathematics questions, released with both generation code and pre-generated questions. This dataset provides an additional measure of algebraic generalization abilities.

### 6.2 RESULTS BASED ON PROMPTING

We first conduct experiments by prompting GPT-4-Turbo-0125[7] to achieve three reasoning modes: Chain of Thought (CoT), Slow Thinking with Dynamic Workflow, and Hybrid Thinking across four benchmark datasets. Table 1 shows that slow thinking with dynamic workflow significantly outperforms CoT by 22.4% on average across four benchmarks. It also reveals that Hybrid Thinking achieves the best accuracy on three datasets BBH, MATH and GameOf24. Notably, both Slow

---

[7]https://platform.openai.com/docs/models. A full list of prompts can be found in Appendix C.

Table 3: Performance comparison of the original Llama-3-8B-Instruct model and the Llama-3-8B-Instruct after our hybrid thinking tuning. We show the accuracy (%) of the model using CoT v.s. slow thinking (with dynamic workflow) and Hybrid Thinking approaches proposed by us. The Fast/Slow indicates the ratio of Fast and Slow Thinking contributions in the Hybrid approach. Results are derived from the full test set in BBH, MATH, DeepMind Math and GameOf24.

| Methods | BBH | MATH | DeepMind Math | GameOf24 | Avg. |
|---|---|---|---|---|---|
| **Llama-3-8B-Instruct (Original)** | | | | | |
| CoT | 51.7 | 30.0 | 18.6 | 2.7 | 25.8 |
| **Llama-3-8B-Instruct (After Hybrid Thinking Tuning)** | | | | | |
| CoT (Fast Think.) | 58.5 (+6.8) | 37.0 (+7.0) | 34.2 (+15.6) | 5.1 (+2.4) | 33.7 (+7.9) |
| Slow Think. | 61.2 (+9.5) | 37.8 (+7.8) | **48.8** (+30.2) | 15.4 (+12.7) | 40.8 (+15.0) |
| Hybrid Think. | **62.3** (+10.6) | **40.2** (+10.2) | 41.7 (+23.1) | **16.0** (+13.3) | 40.5 (+14.7) |

Table 4: Average number of inference tokens of the original Llama-3-8B-Instruct model and the Llama-3-8B-Instruct after our hybrid thinking tuning on various datasets. Performance is reported in Table 3.

| Methods | BBH | MATH | DeepMind Math | GameOf24 | Avg. Tokens |
|---|---|---|---|---|---|
| **Llama-3-8B-Instruct (Original)** | | | | | |
| CoT | 356 | 496 | 359 | 510 | 430.2 |
| **Llama-3-8B-Instruct (After Hybrid Thinking Tuning)** | | | | | |
| CoT (Fast Think.) | 720 | 985 | 770 | 1384 | 964.7 |
| Slow Think. | 3901 | 5743 | 4395 | 6714 | 5188.2 |
| Hybrid Think. | 2521 | 4414 | 2577 | 6371 | 3970.7 |

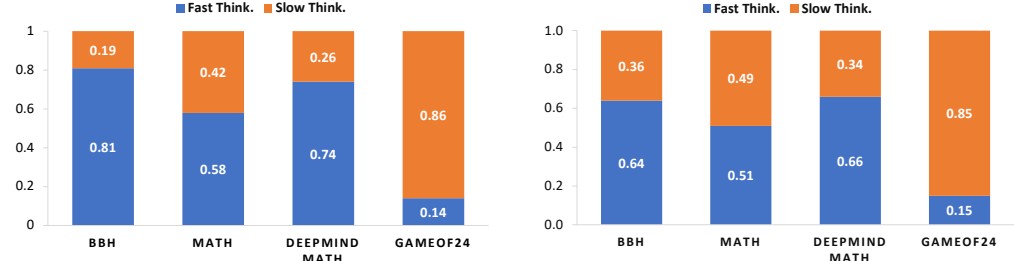

Figure 5: Proportion of fast thinking (CoT) and slow thinking (dynamic workflow) applied in hybrid thinking across four datasets. The left is GPT-4-Turbo (performance is shown in Table 1), while the right is Llama-3-8B-Instruct after our hybrid thinking tuning (Table 3).

Thinking and Hybrid Thinking consistently outperform CoT across all datasets, with the most dramatic improvements seen in GameOf24, where gains are 61.0% and 62.7% respectively.

Table 2 illustrates the average number of inference tokens used by each method. CoT consistently used the fewest tokens (average 577.8), while Slow Thinking required the most (4432.0 on average). Hybrid Thinking struck a balance with an average of 3105.5 tokens. A clear trade-off emerged between computational efficiency and performance, with CoT using the fewest tokens but achieving the lowest accuracy. Hybrid Thinking demonstrated a good balance, achieving high accuracy with moderate token usage. These findings suggest that incorporating dynamic workflows and combining fast and slow thinking processes can enhance the reasoning capabilities of LLMs, with Hybrid Thinking emerging as a particularly promising approach.

Table 5: Accuracy comparison of the original Llama-3-8B-Instruct model and the Llama-3-8B-Instruct after our hybrid thinking tuning on different domains of the MATH dataset. "Count. and Prob." and "Inter. Algebra" represents "Counting and Probability" and "Intermediate Algebra".

| MATH Subsets | Llama-3-8B-Ins. | Llama-3-8B-Ins. (After Hybrid Thinking Tuning) | | | |
| | CoT | CoT (Fast Think.) | Slow Think. | Hybrid Think. | Fast/Slow |
| --- | --- | --- | --- | --- | --- |
| Prealgebra | 43.2% | 58.9% | 59.7% | **63.3%** | 0.69/0.31 |
| Algebra | 30.2% | 53.6% | 52.7% | **56.1%** | 0.68/0.32 |
| Number Theory | 15.0% | 31.1% | 37.6% | **38.0%** | 0.52/0.48 |
| Count. and Prob. | 21.1% | 32.5% | 34.2% | **35.9%** | 0.48/0.52 |
| Geometry | 13.4% | 24.8% | 23.6% | **26.3%** | 0.33/0.67 |
| Precalculus | 12.5% | 22.0% | 21.8% | **24.5%** | 0.35/0.65 |
| Inter. Algebra | 9.1% | 15.6% | 16.3% | **17.3%** | 0.30/0.70 |

## 6.3 RESULTS OF HYBRID THINKING TUNING

We next compare the performance of the original Llama-3-8B-Instruct model and the model after our hybrid thinking tuning. As shown in Table 3, the Llama-3-8B-Instruct model after hybrid thinking tuning significantly outperforms the baseline model on all datasets. Examining the different thinking modes, hybrid thinking consistently provided the best tradeoff between performance and efficiency. Compared to the CoT baseline, hybrid thinking improved accuracy by 10.6%, 10.2%, 23.1% and 13.3% on the BBH, MATH, DeepMind Math and GameOf24 datasets respectively. Interestingly, we also observe that hybrid thinking tuning enhances Llama-3's fast thinking (CoT) performance across all reasoning tasks at the cost of increased model inference tokens.

Table 5 breaks down performance on the MATH dataset into specific subject areas. Again, the Llama-3-8B-Instruct model after hybrid thinking tuning outperforms the original model on all subsets, with gains ranging from 8% on intermediate Algebra to 23% on Number Theory. Hybrid thinking yielded the highest accuracy in each domain, demonstrating its broad applicability.

## 6.4 FAST/SLOW ROUTING ANALYSIS

Figure 5 illustrates the proportion of fast thinking and slow thinking (orange) approaches applied by both models when solving complex problems across the datasets. The GPT-4-Turbo model demonstrates a higher reliance on fast thinking for BBH, DeepMind MATH, and Game of 24 tasks compared with Llama-3-8B-Instruct model. This observation can be attributed to the fact that GPT-4-Turbo's fast thinking (in the form of CoT) is more reliable and effective compared to Llama-3-8B-Instruct. As a result, hybrid thinking in GPT-4-Turbo tends to apply more fast thinking since it is sufficient to achieve a correct solution in many cases. In contrast, Llama-3-8B-Instruct after tuning exhibits a greater reliance on slow thinking strategies, particularly in complex tasks, where fast thinking alone may not yield the desired results. This highlights the importance of hybrid thinking to improve problem-solving efficiency, suggesting that our method can dynamically adjust the optimal balance between fast and slow thinking based on the model's downstream reasoning capabilities.

In summary, the dynamic combination of fast and slow thinking modes greatly enhanced the model's problem-solving capabilities. Our results showcase the potential of hybrid thinking approaches to expand the frontier of what LLMs can achieve on challenging tasks.

## 7 CONCLUSION

This paper introduces a novel framework HDFlow for enhancing the complex problem-solving capabilities of LLMs through hybrid thinking and dynamic workflows. The dynamic workflow mechanism enables LLMs to decompose complex problems into manageable sub-tasks and integrate specialized language models and symbolic reasoning tools, while hybrid thinking strategically engages deeper, multi-step reasoning for challenging problems that exceed the capabilities of fast thinking alone. Extensive experiments demonstrate the significant advantages of our approach, with slow thinking with dynamic workflow greatly outperforming CoT and hybrid thinking achieving the highest overall accuracy by balancing efficiency and performance.

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

## A    TRAINING DETAILS OF HYBRID THINKING TUNING

The Llama-3 models have demonstrated superior performance compared to other models of similar size due to significant enhancements in both pretraining and post-training (Meta, 2024). Therefore, we choose the Llama-3-8B-Instruct model as the foundation model for our hybrid thinking tuning experiments due to its SOTA performance on benchmarks. Specifically, The Llama-3-8B-Instruct model was fine-tuned using 8 A100 GPUs with bf16 precision[8]. The training utilized a global batch size of 128, spanning 4 epochs. The model employed the AdamW optimizer of a learning rate of 2.0e-5, with a maximum sequence length of 4096 tokens and a maximum of 2048 new tokens generated.

## B    DYNAMIC WORKFLOW EXAMPLE SOLUTION

In this section, we demonstrate how our dynamic workflow approach works by presenting an example solution to a reasoning problem. We also present the solution provided by the OpenAI o1-preview model to compare with our approach. We can see the solution produced by our approach contains many sub-goals (sub-tasks), which is very similar to the o1-preview solution. Our approach also leverages the symbolic tool - Python to enhance the accuracy of the final answer. The reasoning problem is generated using our data synthesis method.

### B.1    REASONING PROBLEM:

```
An encrypted Morse code message contains a list of three items that were
    crucial to a historical event. The Morse code is: '- . .- / -.-. ---
    ..-. ..-. . . / ... ..- -- . .- .-.'. Decode this Morse code message
    into English text, keeping in mind the standard Morse code
    conventions where each sequence of dots (.) and dashes (-)
    corresponds to a letter, and spaces separate words.
```

### B.2    SOLUTION USING DYNAMIC WORKFLOW:

**Stage 1. Problem Reflection**

```
===== Meta-Expert@REFLECTION =====
### Problem Reflection start ###

**Task Description**: Decode a given Morse code message into English text
    .

**Elements of the Task**:
1. The message is encoded in Morse code, which uses a series of dots (.)
    and dashes (-) to represent letters and spaces to separate words.
2. Decoding involves translating each Morse code sequence into its
    corresponding English letter or symbol.
3. The Morse code provided is a list of three items related to a
    historical event, which implies the output should be meaningful and
    correctly formatted.

**Sub-tasks Identification**:
1. **Morse Code Dictionary Creation**: Establish a dictionary in Python
    where keys are Morse code sequences (dots and dashes) and values are
    corresponding English letters. This will facilitate the translation
    process.
2. **Parsing the Morse Code**: Split the input Morse code string into
    individual Morse code sequences for letters and words. This involves
    handling spaces correctly to differentiate between letters and word
    boundaries.
```

---

[8]We adopt LitGPT (AI, 2023) in our model training.

```
3. **Translation Process**: For each Morse code sequence obtained from
   the parsing step, use the Morse code dictionary to translate it into
   the corresponding English letter.
4. **Reconstruction of Message**: Combine the translated letters into
   words and then into the complete message. This step ensures that the
   output is a readable English sentence.
5. **Output Formatting**: Format the final output to ensure it is clear
   and presentable, adhering to proper English syntax and spacing.

### Problem Reflection end ###
```

**Stage 2. Workflow Design**

```
===== Meta-Expert@EXPERTS_DESIGN =====
### Specialized Experts Design start ###

**Morse Code Dictionary Creation Expert**: Specializes in creating a
   comprehensive dictionary that maps Morse code symbols to English
   letters. This expert will ensure that all Morse code sequences are
   accurately represented with their corresponding English characters,
   which is crucial for the decoding process. Expert card (in JSON
   format): {"Name": "Morse␣Code␣Dictionary␣Creation␣Expert", "
   Expert_Type": "LLM", "Input_Type": "None", "Output_Type": "Dict[str,␣
   str]"}

**Morse Code Parsing Expert**: Skilled in breaking down the Morse code
   string into individual Morse sequences for letters and words. This
   expert will handle the input Morse code string, correctly identifying
    the separation between letters and words, which is essential for
   accurate translation. Expert card (in JSON format): {"Name": "Morse␣
   Code␣Parsing␣Expert", "Expert_Type": "LLM", "Input_Type": "str", "
   Output_Type": "List[List[str]]"}

**Python Expert of Translation**: Utilizes Python to map Morse code
   sequences to English letters using the dictionary provided by the
   Morse Code Dictionary Creation Expert. This expert will translate
   each Morse code sequence into the corresponding English letter,
   ensuring that the translation is accurate and adheres to the Morse
   code standards. Expert card (in JSON format): {"Name": "Python␣Expert
   ␣of␣Translation", "Expert_Type": "Tool", "Input_Type": "List[List[str
   ]],␣Dict[str,␣str]", "Output_Type": "List[str]"}

**Message Reconstruction Expert**: Specializes in reconstructing the
   translated English letters into coherent words and sentences. This
   expert ensures that the translated letters are correctly combined to
   form meaningful words and that these words are arranged into a
   readable and grammatically correct sentence. Expert card (in JSON
   format): {"Name": "Message␣Reconstruction␣Expert", "Expert_Type": "
   LLM", "Input_Type": "List[str]", "Output_Type": "str"}

**Final Review and Presentation Expert**: Responsible for reviewing the
   reconstructed message to ensure it meets the clarity and accuracy
   standards. This expert will also format the final message for
   presentation, ensuring that it is clear, concise, and ready for
   delivery as the solution to the problem. Expert card (in JSON format)
   : {"Name": "Final␣Review␣and␣Presentation␣Expert", "Expert_Type": "
   LLM", "Input_Type": "str", "Output_Type": "str"}

### Specialized Experts Design end ###
```

**Stage 3. Graph Construction and Execution**

```
===== Morse Code Dictionary Creation Expert =====
### Approach Explanation Start ###
```

```
To solve the assigned sub-task of creating a comprehensive Morse code
    dictionary, I will map each Morse code symbol (comprising dots '.'
    and dashes '-') to its corresponding English letter. This dictionary
    is essential for the subsequent decoding process, as it allows for
    the translation of Morse code sequences into readable English text.

**Steps to Create the Morse Code Dictionary**:
1. **Identify Morse Code Symbols**: List all Morse code symbols for
    English alphabet letters (A-Z) and numerals (0-9) if necessary.
2. **Map to English Letters**: Each Morse code symbol will be associated
    with its corresponding English letter. This mapping is based on the
    standard Morse code conventions.
3. **Dictionary Format**: The dictionary will be structured in Python
    dictionary format, where keys are Morse code symbols and values are
    the corresponding English letters.

This dictionary will be used by the next expert in the workflow, the
    Morse Code Parsing Expert, who will parse the Morse code message into
    individual symbols using this dictionary for translation.

### Approach Explanation End ###

### My Final Output Start ###
{
    "A": ".-",
    "B": "-...",
    "C": "-.-.",
    "D": "-..",
    "E": ".",
    "F": "..-.",
    "G": "--.",
    "H": "....",
    "I": "..",
    "J": ".---",
    "K": "-.-",
    "L": ".-..",
    "M": "--",
    "N": "-.",
    "O": "---",
    "P": ".--.",
    "Q": "--.-",
    "R": ".-.",
    "S": "...",
    "T": "-",
    "U": "..-",
    "V": "...-",
    "W": ".--",
    "X": "-..-",
    "Y": "-.--",
    "Z": "--..",
    "1": ".----",
    "2": "..---",
    "3": "...--",
    "4": "....-",
    "5": ".....",
    "6": "-....",
    "7": "--...",
    "8": "---..",
    "9": "----.",
    "0": "-----"
}
### My Final Output End ###

===== Morse Code Parsing Expert =====
```

```
### Approach Explanation Start ###

To solve the task of parsing the Morse code string into individual Morse
    sequences for letters and words, I will follow these steps:

1. **Understanding Morse Code Structure**: Morse code uses dots (.) and
    dashes (-) to represent letters, and spaces are used to separate
    different Morse sequences. A single space indicates a separation
    between letters within the same word, and a sequence of three spaces
    (represented as "␣/␣") indicates a separation between words.

2. **Splitting the Input String**: The input Morse code string will be
    split based on the "␣/␣" sequence to differentiate between words.
    This results in a list where each element is a Morse code sequence
    representing a word.

3. **Further Splitting for Letters**: Each word obtained from the
    previous step will be further split using a single space as the
    delimiter. This will break down the word into individual Morse code
    sequences, each representing a letter.

4. **Constructing the Output**: The final output will be a list of lists,
    where each sublist represents a word and contains Morse code
    sequences for each letter in that word.

This structured approach ensures that the Morse code is accurately parsed
    into its constituent letters and words, preparing it for translation
    in the subsequent steps of the workflow.

### Approach Explanation End ###

### My Final Output Start ###
[['- . .-', '-.-. --- ..-. ..-. . .', '... ..- --. .- .-.']]
### My Final Output End ###

===== Python Expert of Translation =====
### Python Code Start ###

def translate_morse_code(morse_words, morse_dict):
    translated_words = []
    for word in morse_words:
        translated_word = []
        for letter in word.split():
            # Using next() with a default value to avoid StopIteration
            translated_letter = next((key for key, value in morse_dict.
                items() if value == letter), '?')
            translated_word.append(translated_letter)
        translated_words.append(''.join(translated_word))
    return translated_words

morse_words = [['- . .-', '-.-. --- ..-. ..-. . .', '... ..- --. .-
    .-.']]
morse_dict = {
    "A": ".-",
    "B": "-...",
    "C": "-.-.",
    "D": "-..",
    "E": ".",
    "F": "..-.",
    "G": "--.",
    "H": "....",
    "I": "..",
    "J": ".---",
    "K": "-.-",
    "L": ".-..",
```

```
    "M": "--",
    "N": "-.",
    "O": "---",
    "P": ".--.",
    "Q": "--.-",
    "R": ".-.",
    "S": "...",
    "T": "-",
    "U": "..-",
    "V": "...-",
    "W": ".--",
    "X": "-..-",
    "Y": "-.--",
    "Z": "--..",
    "1": ".----",
    "2": "..---",
    "3": "...--",
    "4": "....-",
    "5": ".....",
    "6": "-....",
    "7": "--...",
    "8": "---..",
    "9": "----.",
    "0": "-----"
}

for words in morse_words:
    translated = translate_morse_code(words, morse_dict)
    print(translated)

### Python Code End ###

### Python Execution Result Start ###
['TEA', 'COFFEE', 'SUGAR']
### Python Execution Result End ###

===== Message Reconstruction Expert =====
### My Approach Explanation Start ###

As the Message Reconstruction Expert, my primary role is to take the
    translated English words provided by the Python Expert of Translation
     and construct them into a coherent and grammatically correct
    sentence. The translated words are 'TEA', 'COFFEE', and 'SUGAR'.
    These words represent items, and the context suggests they are a list
     of crucial items related to a historical event.

Given this context, my task is to format these words into a readable
    English sentence that clearly communicates the items as a list. The
    appropriate format for presenting such a list in English would
    typically involve separating the items with commas and using the
    conjunction "and" before the last item. This ensures the sentence is
    both grammatically correct and easy to understand.

### My Approach Explanation End ###

### My Final Output Start ###
Tea, coffee, and sugar
### My Final Output End ###

===== Final Review and Presentation Expert =====
### Explanation of Approach Start ###

As the Final Review and Presentation Expert, my primary role is to ensure
     that the reconstructed message from the Message Reconstruction
    Expert is clear, accurate, and well-presented. The message provided
```

```
      is "Tea, coffee, and sugar," which lists three items crucial to a
      historical event.

**Steps in my approach:**
1. **Review Accuracy**: Check if the message accurately reflects the
   translated Morse code. The previous experts have translated and
   reconstructed the message correctly as "Tea, coffee, and sugar" from
   the Morse code '- . .- / -.-. --- ..-. ..-. . . / ... ..- --. .-
   .-.'.
2. **Review Clarity**: Ensure the message is clear and understandable.
   The use of commas and the conjunction "and" appropriately lists the
   items, making the sentence coherent and easy to read.
3. **Presentation**: Format the message to ensure it is ready for
   presentation. This involves checking for any grammatical errors,
   proper capitalization, and punctuation.

Given the outputs from the previous experts, the message "Tea, coffee,
   and sugar" is already well-constructed. My task focuses on confirming
    these aspects and presenting the final output.

### Explanation of Approach End ###

### My Final Output Start ###
Tea, coffee, and sugar
### My Final Output End ###

===== Meta-Expert@FINAL_JUDGEMENT =====
### Analysis Start ###

**Problem Recap**: The task was to decode a Morse code message that lists
    three items crucial to a historical event. The Morse code provided
    was '- . .- / -.-. --- ..-. ..-. . . / ... ..- --. .- .-.'.

**Workflow Review**:
1. **Morse Code Dictionary Creation**: The dictionary was correctly
   established, mapping each Morse code sequence to its corresponding
   English letter. This is foundational for accurate translation.

2. **Morse Code Parsing**: The parsing expert successfully identified and
    separated the Morse code into three distinct word sequences, which
   is crucial for correct translation.

3. **Translation Process**: Using the dictionary, the translation expert
   accurately converted the Morse code sequences into the English words
   'TEA', 'COFFEE', and 'SUGAR'. This step was executed correctly,
   adhering to Morse code standards.

4. **Message Reconstruction**: The reconstruction expert transformed the
   list of words into a coherent and grammatically correct English
   phrase: "Tea, coffee, and sugar". This step was crucial in making the
    output readable and presentable.

5. **Final Review and Presentation**: The final output "Tea, coffee, and
   sugar" was reviewed and presented clearly. The expert ensured that
   the output was concise and correctly formatted, suitable for
   answering the problem statement.

**Evaluation of Accuracy and Completeness**:
- The translation of Morse code to English was accurate based on the
   provided dictionary.
- The final message is meaningful and directly corresponds to the Morse
   code given.
- The output is formatted in a way that is easy to understand and
   relevant to the context of a list of items.
```

```
**Conclusion**:
The workflow from dictionary creation to final presentation was executed
    seamlessly, with each expert fulfilling their role effectively. The
    final output accurately reflects the decoded message from the Morse
    code, presented in a clear and concise manner.

### Analysis End ###

FINAL EVALAUTION: YES
```

## B.3  SOLUTION USING OPENAI O1-PREVIEW

**Hidden Thought Process (thought for 12 seconds)**

```
===== Parsing Morse code =====
I'm separating the Morse code message into three words. Each word is then
    further divided into its constituent letters for decoding.

===== Parsing Morse code =====
In this chunk, the task involves parsing Morse code by handling letter
    concatenation within words and word separation through slashes. Morse
    code typically separates letters with spaces and words with slashes.

===== Breaking down Morse code =====
I'm looking at Morse code to decode - . .-, mapping it to letters. This
    makes me think about interpreting symbols as letters.

===== Considering Morse code decoding =====
I'm thinking through the method of reversing Morse code, focusing on
    identifying letter partitions and parsing two-letter words to
    determine valid Morse code sequences, examining all possible
    partitions of the code.

===== Identifying letter patterns =====
I'm working through different Morse code partitions to isolate letter
    patterns like T E A and T U. This involves analyzing various
    combinations of Morse code symbols.

===== Decoding Morse code =====
It's interesting to see that the Morse code was parsed as -.-. --- ..-.
    ..-. . . for 'COFFIE' and 'T E A' as the most plausible word. This
    suggests a detailed analysis of each symbol is crucial for accurate
    translation.

===== Decoding Morse code =====
I'm working through the Morse code sequence, breaking down each symbol to
    reveal letters like C, O, F, F, E, E, S, U, G, and ?.
```

**Final solution**

```
To decode the Morse code message '- . .- / -.-. --- ..-. ..-. . . / ...
    ..- --. .- .-.', we'll break it down word by word and letter by
    letter.

### Morse Code Reference Table:

| Letter | Morse Code |
| --- | --- |
| A | .- |
| B | -... |
| C | -.-. |
| D | -.. |
| E | . |
```

```
| F | ..-. |
| G | --. |
| H | .... |
| I | .. |
| J | .--- |
| K | -.- |
| L | .-.. |
| M | -- |
| N | -. |
| O | --- |
| P | .--. |
| Q | --.- |
| R | .-. |
| S | ... |
| T | - |
| U | ..- |
| V | ...- |
| W | .-- |
| X | -..- |
| Y | -.-- |
| Z | --.. |
```

### Decoding Process:

**First Word: `- . .-`**

* `-` : **T**
* `.` : **E**
* `.-` : **A**

Combined: **TEA**

**Second Word: `-.-. --- ..-. ..-. . .`**

* `-.-.` : **C**
* `---` : **O**
* `..-.` : **F**
* `..-.` : **F**
* `.` : **E**
* `.` : **E**

Combined: **COFFEE**

**Third Word: `... ..- --. .- .-.`**

* `...` : **S**
* `..-` : **U**
* `--.` : **G**
* `.-` : **A**
* `.-.` : **R**

Combined: **SUGAR**

### Final Decoded Message:

**TEA COFFEE SUGAR**

* * *

These three items—**tea**, **coffee**, and **sugar** – were crucial
    commodities in historical trade events, such as the Boston Tea Party
    and the global trade networks during the colonial era.

**Answer:**

```
TEA COFFEE SUGAR
```

## C   PROMPTS USED IN OUR DYNAMIC WORKFLOW APPROACH

Note that placeholders enclosed in {} are used within the prompts to represent variables that will be substituted with actual values at LLM query time.

---

**Problem Reflection Prompt**

### Problem Statement Start ###
{task_problem}
### Problem Statement End ###
You are an exceptionally capable Meta-Expert, possessing a unique capability for conducting problem reflection. Your primary function involves receiving the above problem query, which you must methodically decompose into smaller, more manageable sub-tasks (including sub-tasks that can solved by implementing Python functions). When designing the solution, you should think about its generalizability. A robust solution can tackle a similar range of problems effectively with minor adaptations. This decomposition will later facilitate the creation of a team of specialized experts, enabling efficient collaboration of experts to address and solve the above problem. When breaking down into sub-tasks, it is crucial to:
1. Ensure Sequential Logic: Arrange the sub-tasks in a logical, sequential order that facilitates a smooth workflow from start to finish.
2. Avoid Overlap: Each sub-task must be distinct, with no duplication of efforts across the tasks, ensuring efficient allocation of expertise.
3. Pursue Optimal Decomposition: Ensure sub-tasks are sufficiently defined to be tackled effectively. Maintain a manageable number of specific sub-tasks, facilitating easier coordination and management. In particular, please conduct the "Problem Reflection" for the given problem: Reflect on the problem, and describe it in your own words, in bullet points. Analyze how you can decompose the problem into smaller, more manageable sub-tasks. Note that you can integrate Python-driven sub-tasks by implementing and running modular Python code if necessary. Pay attention to small details, nuances, notes and examples in the problem description.

---

**Experts Design Prompt**

### Problem Statement Start ###
{task_problem}
### Problem Statement End ###
### Problem Reflection Start ###
{problem_reflection}
### Problem Reflection End ###
You are an extremely powerful Meta-Expert with the unique ability to design a team of specialized experts and arrange those experts through a workflow to tackle and solve the above problem. Based on the above problem statement and its reflection analysis, please design a team of experts and orchestrate those experts to effectively address and solve the above problem.
In particular, you are to do "Specialized Experts Design":
- Design a list of subject-matter experts (SMEs) including, but not limited to, Essayist Expert, Python Expert, Linguistic Analyst, Mathematician, Data Scientist, and various other Analysts. Each expert is only to perform one specific sub-task, such as processing data, making decisions, or utilizing Python tools.
- Arrange the experts to operate in a sequential workflow, meaning each expert's output becomes the input for the next, progressively moving towards the final answer. Avoid redundancy of functions across experts.
- Assign unique names to each expert and provide an clear description of their specific skills, knowledge, and the sub-tasks they are going to perform. Ensure the expert description is comprehensive and self-contained that encapsulates all important information and details from **Sub-tasks Identification**.
- For sub-tasks involving logical reasoning, mathematical operations, data structure manipulation, or programming-related challenges, you can outline strategic approaches and delegate the specifics of implementation to the Python expert (Tool). The Python expert will translate the instructions into code, execute it, and return the results. You can include multiple Python experts if needed. Please provide explicit implementation instructions to the Python expert(s).
- Conclude each expert's description with a name card in JSON format, summarizing key attributes. Specify the type of each expert as either 'LLM' for those based on Large Language Model or 'Tool' for those utilizing Python tools.
- The final expert should be responsible for reviewing the findings of previous experts and then generating the final answer to the problem.

**Execution Prompt of Experts Initiated from LLM**

### Problem Statement Start ###
{original_problem}
### Problem Statement End ###
### Problem Reflection Start ###
{problem_reflection}
### Problem Reflection End ###
Please act as {name}. Your role: {role} You are part of a specialized expert team. You are designed to accomplish a sub-task and collaborate with other experts through a workflow graph to solve the above problem.
The expert team operates based on the following design:
### Experts Design Start ###
{experts_design}
### Experts Design End ###
Each expert, including you, is responsible for a specific sub-task. The workflow is structured so that each expert's output becomes the input for the next, progressively moving towards the final answer. The process should be thought of as sequential steps, where you contribute towards the solution based on the outputs from the previous experts.{data_type_instruction} You can think step by step if necessary.
The results from the preceding experts are as follows:
### Experts' Results Start ###
*input_data*
### Experts' Results End ###
Please provide a brief explanation of your approach to solving the assigned sub-task. After your explanation, clearly indicate your final output as follows:
### My Final Output Start ###
[Your final answer here]
### My Final Output End ###

---

**Execution Prompt of Experts initiated from Symbolic Engine**

### Problem Statement Start ###
{original_problem}
### Problem Statement End ###
### Problem Reflection Start ###
{problem_reflection}
### Problem Reflection End ###
Please act as {name}. Your role: {role} You are a specialized Python expert among a team of experts. You are designed to write Python code to accomplish a sub-task and collaborate with other experts through a workflow graph to solve the above problem.
The expert team operates based on the following design:
### Experts Design Start ###
{experts_design}
### Experts Design End ###
Each expert, including you, is responsible for a specific sub-task. The workflow is structured so that each expert's output becomes the input for the next, progressively moving towards the final answer. You should take the previous expert's output as input, write the Python code, execute the code, and send the output to the next expert.
The results from the preceding experts are as follows:
### Experts' Results Start ###
*input_data*
### Experts' Results End ###
Please write the Python code that takes input in {input_type} and return output in {output_type}.
Guidelines: - Make sure the code includes all the necessary module imports, properly initialize the variables, and address the problem requirements. - The code needs to be self-contained, and executable as-is. Output only code, without any explanations or comments.
The code output must follow this structure:

```python
def f1(...):
    ...
    return ...

def f2(...):
    ...
    return ...
...

if __name__ == "__main__":
    ...
```

*how_to_read_input*
The output should be printed without additional words using the 'print()' method.
Answer:

```python
```

---

**Verification Prompt**

### Problem Statement Start ###
{task_problem}
### Problem Statement End ###
### Problem Reflection Start ###
{problem_reflection}
### Problem Reflection End ###
**Experts Design:** - Based on the problem reflection, a team of experts has been designed and organized through a workflow to tackle and solve the problem described above. - Experts are designed to operate in a sequential workflow, meaning each expert's output becomes the input for the next, progressively moving towards the final answer. - The final expert is responsible for reviewing the findings of previous experts and then generating the final answer to the problem.
Here is a description of the experts' roles and the workflow structure:
### Experts Design Start ###
{experts_design}
### Experts Design End ###
Based on the workflow design, the experts have provided the following results:
### Experts' Results Start ###
{experts_results}
### Experts' Results End ###
Given the described workflow design and the results produced by the experts, your task is to evaluate whether the final output of the "{final_expert}" successfully and correctly solves the problem presented.
Please provide your analysis and then conclude your evaluation by stating 'FINAL EVALUATION: YES' or 'FINAL EVALUATION: NO'.

# D    DATA SYNTHESIS OF REASONING PROBLEMS

**Data Synthesis Prompt 1**

Please develop 10 new and diverse reasoning tasks, one per line, inspired by but distinct from the following 10 example reasoning tasks:
{example_tasks}
Guidelines for task creation:
- Ensure each new task is distinctly different from the example tasks provided; avoid mere variations.
- Clearly and accurately define each task, making its objective and scope explicit.
- Design tasks that yield deterministic answers, facilitating the creation of single, definitive standard answers for subsequent problems derived from these tasks. This helps straightforward evaluation of correctness.
- Target a moderate to hard difficulty level for each task, requiring thorough analysis and in-depth reasoning to solve.

**Data Synthesis Prompt 2**

Please develop 10 new and diverse puzzle tasks, one per line, to test various reasoning abilities.
Guidance:
- Each new puzzle task should clearly and accurately describe what the task is.
- Design puzzle tasks that yield deterministic answers, facilitating the creation of single, definitive standard answers for subsequent problems derived from these tasks. This helps straightforward evaluation of correctness.
- Puzzle tasks should have a moderate to hard difficulty level - they should require thorough analysis and in-depth reasoning to work through.

**Problem Validation Prompt**

### Problem Start ###
{problem}
### Problem End ###
Your task is to verify whether the above problem is a valid reasoning problem or not.
Valid Criteria:
- It is clear and unambiguous (NO multiple interpretations).
- It provides all necessary information required to solve the problem.
- The problem is logically structured so that it can be approached through reasoning skills. It does not depend on subjective judgments or opinions.
- The problem is solvable and has one single, definitive correct answer that can be derived through reasoning.
- There are no internal contradictions or conflicts in the problem.
Please provide a concise analysis and then output '## VALID ##' or '## INVALID ##'. Next, if it is invalid, please rewrite it into a new valid reasoning problem following the format below. Make sure the new problem is challenging enough.
### New Valid Problem Start ###
[new problem]
### New Valid Problem End ###

