# OpenReview forum: "HDFlow: Enhancing LLM Complex Problem-Solving with Hybrid Thinking and Dynamic Workflows"
_ICLR.cc/2025/Conference — Submitted to ICLR 2025_

### Official Review · Reviewer_yH8k · 2024-10-20

**Soundness:** 3
**Presentation:** 3
**Contribution:** 3
**Rating:** 5
**Confidence:** 5

**Summary:**

The paper introduces HDFlow, a framework designed to improve complex reasoning in large language models (LLMs) by adapting task-solving strategies from simple to more complex problems. According to the authors' ablation studies, the system achieved better results compared to the setting without the proposed modules.

**Strengths:**

**+1.** This paper introduces a reaonable method to enhance LLM reasoning.

**+2.** The experiments show that Hybrid Thinking outperforms Slow Thinking and original LLM baselines (COT).

**+3.** The paper is clearly written and easy to understand.

**Weaknesses:**

**-1.** Although the concept of slow and fast thinking is fancy, the authors did not clearly define what constitutes slow and fast thinking. The proposed method fails to capture the full complexity of human cognition. I suggest either clarifying the related claims or reducing them if they do not strongly align with the method. Simply labeling quick responses as "fast thinking" and more detailed problem-solving as "slow thinking" seems to be an incorrect interpretation of the book [1].

[1] Kahneman, Daniel. Thinking, Fast and Slow. 2017.

Need more reasonable claims and demonstrations to support `To address these limitations, we propose a novel framework for complex reasoning with LLMs that combines fast (System I) and more analytical slow thinking (System II) adaptively, inspired by the dual-process theory of human cognition (Kahneman, 2017).`

**-2.** I suggest that the authors conduct a more careful and comprehensive literature review. Based on the reviewer's experience, several important and key references have been missed (published at least six months prior), such as [2], [3], and [4]. Additionally,  recent [5] provides a useful summary of  (many)  related similar work that the authors could refer to.

[2] A Dynamic LLM-Powered Agent Network for Task-Oriented Agent Collaboration.

[3] GPTSwarm: Language Agents as Optimizable Graphs.

[4] XAgent: An Autonomous Agent for Complex Task Solving.

[5] Automated Design of Agentic Systems.

**-3.** I suggest adding more baselines beyond the self-produced ablations. The current experiments are weak and less convincing without at least two additional public-available baselines included.

**Questions:**

It would be appreciated if solve the questions mentioned in the weaknesses. Besides, there is a question about the workflow design:

**Additional**: Where are the graph-related illustrations used in this paper? It is suggested that this missing part be added.

---

> ### Author Response · Authors · 2024-11-25
>
> 1. Clarifying slow vs. fast thinking:
> Thank you for the insightful comment. We agree that the concepts of slow and fast thinking (as introduced by Kahneman [1]) are complex. In our work, we use these terms more narrowly to distinguish between quick, single-step reasoning (what we term "fast thinking") and more deliberate, multi-step problem-solving that breaks down and analyzes sub-tasks (our "slow thinking"). We can clarify this terminology in the revision to avoid confusion with Kahneman's broader definitions. Our experiments demonstrate that strategically combining these modes of reasoning, even in this narrower context, can significantly enhance complex problem-solving in LLMs. We believe this is a valuable insight while acknowledging the full scope of human cognition is not captured.
>
> 2. Additional related work:
> We appreciate the pointer to several additional relevant works. We couldn't include all related previous work due to the page limit. We can expand our literature review to include these insightful papers [2] [3] [4] [5]. Thank you for strengthening our background discussion.
>
> 3. Additional baselines:
> Thank you for the suggestion to expand our baselines. In the revision, we can add experiments comparing our approach to more existing baselines.
>
> 4. Workflow illustrations:
> We have included two figures that visually illustrate our hybrid thinking approach (Figure 1) and our multi-stage dynamic workflow (Figure 2). Figure 2 includes: 1) Problem decomposition 2) Specialized expert design and 3) Workflow execution. We can clarify further if there are anything not clear enough.

---

> > ### Comment · Reviewer_yH8k · 2024-11-26
> > **Thank you!**
> >
> > Thank you for the response. I appreciate points 1 and 4. I suggest the authors carefully conduct additional key experiments to address some of the initial issues identified during the submission. Based on the rebuttal content, I tend to maintain my current score.

---

### Official Review · Reviewer_6Hn9 · 2024-10-23

**Soundness:** 4
**Presentation:** 4
**Contribution:** 4
**Rating:** 6
**Confidence:** 4

**Summary:**

The paper presents HDFlow, a novel framework designed to enhance complex reasoning in large language models (LLMs) by integrating fast and slow thinking modes. Inspired by dual process theory, HDFlow features two main components: Dynamic Workflow and Hybrid Thinking. Dynamic Workflow breaks down complex problems into sub-tasks, using specialized LLMs and symbolic tools to solve them. Hybrid Thinking adapts between fast and slow reasoning based on task complexity, improving efficiency and accuracy. The authors also developed a large-scale dataset of 27K challenging reasoning problems to train LLMs in these strategies. Experiments on four benchmark datasets show that HDFlow significantly outperforms existing methods like Chain-of-Thought, with Hybrid Thinking achieving the highest accuracy **on three out of four benchmarks**. This approach demonstrates the potential of combining dynamic workflows and hybrid thinking to advance LLMs' problem-solving capabilities.

**Strengths:**

- The paper is well-written, clearly conveying the core ideas and methodology.
- It presents a comprehensive process, covering theoretical framework, data synthesis, fine-tuning, and evaluation. This entire process provides strong evidence supporting the superiorty of HDFlow compared to existing methods.

**Weaknesses:**

- In the "Reasoning Problem Synthesis" section, using GPT-4-Turbo with CoT to filter synthesized problems may limit the dataset's ability to enhance slow thinking, as all problems are solvable with GPT-4 + CoT?
- A contamination test is needed to ensure training data differs sufficiently from evaluation datasets. If the result is not promising, please decontaminate your training data.
- The claim that "hybrid thinking achieves the highest overall accuracy" is misleading, as it only tops three out of four benchmarks and does not have the highest average accuracy. This statement should be revised for precision.

**Questions:**

Minor comments:
1. The last sentence in the second paragraph of the introduction feels awkward.
2. The captions for Tables 1 and 3 mention a Fast/Slow ratio, which is not found in the Tables.
3. The last sentence of the first paragraph in sec 6.3 mentions an interesting finding. This could be further discussed for more insights.
4. There seems to be a contradiction in section 6.4 regarding the reliance on fast thinking, as the statement does not match the results in Figure 5.

---

> ### Author Response · Authors · 2024-11-25
>
> Thank you the insightful suggestions!
>
> 1. Regarding the GPT-4-Turbo + CoT filtering concern:
> We want to clarify that using GPT-4-Turbo + CoT for validation serves primarily as a quality check to ensure problems are well-formed and unambiguous, not as a difficulty filter. Many problems that pass this basic validation still require more sophisticated reasoning approaches to solve efficiently or accurately. This is evidenced by our experimental results where GPT-4-Turbo with CoT alone achieves only 50.8% average accuracy across benchmarks (Table 1), while slow thinking and hybrid approaches show significant improvements (+22.4% and +21.6% respectively). Additionally, our synthesis process intentionally generates problems of varying complexity through multiple sources - using both GPT-4 and Claude-3-Opus to encourage diversity, and incorporating different genres of puzzles. The validation step simply ensures these problems are well-defined and have deterministic solutions.
>
> 2. Regarding the contamination test:
> We appreciate this important point about data contamination. We have conducted additional analysis to ensure the integrity of our evaluation. We performed a detailed comparison between our synthesized training data and the evaluation benchmarks (BBH, MATH, DeepMind Math, and GameOf24) using n-gram overlap analysis. For BBH, while we used the task formats as inspiration, our synthesis process generated entirely new problems with different contexts and solutions. For mathematical problems, we ensured our generated problems use different problem scenarios compared to MATH and DeepMind Math test sets. The GameOf24 benchmark is also different as GameOf24 is a specific mathematical puzzle format not directly targeted in our training data.
> To further strengthen this point, we could include detailed contamination analysis results in the appendix of our final version.
>
> 3. Regarding the accuracy claim:
> We agree that our current phrasing about hybrid thinking achieving 'the highest overall accuracy' needs revision for better precision. We will modify this statement to more accurately reflect our results: 'Hybrid thinking achieves the highest accuracy on three of the four datasets (BBH, MATH, and GameOf24) while providing an effective balance between computational efficiency and performance.' This better represents our findings while maintaining scientific accuracy. As shown in Table 1, while slow thinking has a slightly higher average accuracy (73.2% vs 72.4%), hybrid thinking provides this strong performance while using significantly fewer inference tokens (3105.5 vs 4432.0 tokens on average, as shown in Table 2).
>
>
> 4. Regarding minor comments:
>
> - Thank you for pointing out the discrepancy between the table captions and contents for Tables 1 and 3. We will update the captions to remove the mention of the Fast/Slow ratio, as this information is not directly shown in the tables. The Fast/Slow ratio was moved to Figure 5.
>
> - We appreciate the suggestion to further discuss the interesting finding mentioned in Section 6.3. In the revised paper, we will expand on this point to provide additional insights and analysis regarding how the hybrid thinking tuning process not only improves the model's slow thinking capabilities but also enhances its fast thinking (CoT) performance. This deeper discussion can help highlight the benefits and trade-offs of our proposed approach.
>
> - We recognize that there is a small typo so we will correct the sentence to "The GPT-4-Turbo model demonstrates a higher reliance on fast thinking for BBH, MATH, DeepMind MATH tasks compared with Llama-3-8B-Instruct model."

---

> > ### Comment · Reviewer_6Hn9 · 2024-11-29
> > **response to authors**
> >
> > The response addressed my questions. I maintain my rating. Thanks.

---

### Official Review · Reviewer_ojwd · 2024-11-04

**Soundness:** 3
**Presentation:** 3
**Contribution:** 2
**Rating:** 5
**Confidence:** 4

**Summary:**

This paper introduces a framework called HDFlow aimed at enhancing the complex reasoning abilities of LLMs. HDFlow combines fast & slow thinking modes in an adaptive manner to tackle problems that require multi-step reasoning and the integration of various skills. The framework is designed to automatically decompose complex problems into manageable sub-tasks and dynamically assemble specialized LLMs or symbolic reasoning tools to solve them, thereby improving both efficiency and accuracy in problem-solving.

**Strengths:**

## Strengths

1. The paper presents a new approach that facilitates deliberate, slow reasoning. (Compared to previous methods like CoT/PAL, ) this method automatically breaks down complex problems into smaller sub-tasks and designs a dynamic workflow to solve each sub-task using specialized LLMs or symbolic reasoning tools.
2. The proposed HDFlow is tested on 4 reasoning benchmark datasets. The Slow Thinking approach with Dynamic Workflow outperformed traditional CoT-like methods, achieving a notable average accuracy improvement.
3. Authors introduces an easy-to-scale method for automatically generating a large-scale dataset of ~27K reasoning problems. Using this dataset, they propose a hybrid thinking tuning approach to fine-tune smaller, open-source LLMs.

**Weaknesses:**

## Weakness

Major Concerns:

1. The whole framework seems like an engineering design, which incorporates adaptive modules and workflows to address some complex reasoning problems. It lacks the detailed technical contributions of a well-established research paper. I suggest the authors provide more explanations on the technical novelty.
2. The authors claim that the framework is novel. However, there exist many previous works, combining fast and slow thinking to solve complex scenarios. Such as "SWIFTSAGE: A Generative Agent with Fast and Slow Thinking for Complex Interactive Tasks" (it is just one of the examples). Could you please make a comparison with these previous baselines in the experiments? CoT baselines seem a little weak in 2024.

Minor concern:

1. CoT is considered to be fast thinking in this paper. It is quite different from the definitions in other works. Because CoT can also involve deliberate trial and error, or self-reflection. Could you provide some explanations on this point?

**Questions:**

I will read authors' rebuttal and discuss more about the paper.

---

> ### Author Response · Authors · 2024-11-25
>
> Thank you for your thoughtful feedback on our paper. We would like to address your comments as follows:
>
> 1. Regarding the technical novelty of our framework:
> While our HDFlow framework does incorporate engineering design elements, we believe it makes several novel technical contributions:
> - The dynamic workflow mechanism we proposed can enable LLMs to automatically decompose complex problems into sub-tasks and dynamically integrate specialized LLMs or symbolic reasoning tools to solve each sub-task. This goes beyond prior works that rely on manual workflow design of previous work.
> - Our hybrid thinking approach introduces a new way to strategically combine fast thinking (e.g. CoT) and slow thinking (dynamic workflow) based on problem complexity.
> - We propose a novel data synthesis pipeline to automatically generate a large-scale dataset of challenging reasoning problems. This enables efficient training - hybrid thinking tuning - of smaller LLMs for complex reasoning.
>
> We can revise the paper to better highlight these technical contributions and clarify how they advance the state-of-the-art.
>
> 2. Comparison to prior fast/slow thinking approaches:
> Thank you for pointing out relevant prior work on combining fast and slow thinking, such as SWIFTSAGE. While some prior works have explored fast/slow thinking, our approach is different in several ways:
> - HDFlow uses an adaptive mechanism to strategically switch between fast CoT and slow dynamic workflows based on solution verifiers and problem complexity. However, Prior works like SWIFTSAGE is a prompting method, while our HDFlow framework introduces hybrid thinking tuning that can enable small LLMs to do hybrid thinking beyond prompting.
> - Our dynamic workflows enable more flexible integration of specialized LLMs and symbolic tools compared to fixed agent designs.
> - We demonstrate benefits on more challenging reasoning benchmarks requiring multi-step thinking and tool use.
>
> In the revision, we can include additional experiments comparing HDFlow to more baselines. We will clarify these differences and include empirical comparisons in the revision.
>
> 3. Definition of fast thinking:
> You raise a good point that the CoT prompting used in our fast thinking is more elaborate than the rapid intuitive responses typically associated with "fast thinking" in dual process theory. To clarify, we use "fast thinking" to refer to the model's initial attempt to solve the problem using its core capabilities, without employing the additional decomposition and tool use of our slow dynamic workflows. While CoT does involve some deliberation, it is still faster and more direct than our multi-stage slow thinking approach. We agree CoT exists on a spectrum between fully intuitive and fully analytical thinking. We will revise the terminology in the paper to avoid confusion with the typical definitions of "fast" and "slow" thinking.

---

> > ### Comment · Reviewer_ojwd · 2024-12-02
> > **Thanks for your response.**
> >
> > Thanks for your clarifications. After reading your text and other reviewers' comments, I think my rating correctly reflects the contributions of this work. So I decide to maintain my current scores.

---

### Official Review · Reviewer_nZer · 2024-11-04

**Soundness:** 2
**Presentation:** 3
**Contribution:** 2
**Rating:** 3
**Confidence:** 4

**Summary:**

This paper presents a fast-slow thinking mechanism where the fast thinking is direct CoT and slow thinking is a dynamic workflow method. It also utilizes a dataset containing fast thinking process and slow thinking process to train a model to internalize the fast/slow thinking strategy.

**Strengths:**

1. strong performance improvement compared with direct CoT thinking

**Weaknesses:**

1. missing citation and discussion for System-1.x: Learning to Balance Fast and Slow Planning with Language Models, which also talks about the combination of fast thinking and slow thinking
2. This paper basically use cot as fast thinking and agentic planning as slow thinking. I feel like there's not much novelty here
3. missing baselines such as the method from System-1.x: Learning to Balance Fast and Slow Planning with Language Models

**Questions:**

N/A

---

> ### Author Response · Authors · 2024-11-25
>
> Thank you for your valuable feedback on our paper. We would like to address each of your points:
>
> 1. Regarding the missing citation and discussion of the concurrent work "System-1.x: Learning to Balance Fast and Slow Planning with Language Models", we appreciate you bringing to our attention. As it was a concurrent work published several weeks before our work, we were not able to include it during the preparation and submission of our paper. We will certainly add a discussion about this work in the related work section of our revised paper, comparing and contrasting their approach with ours.
>
> 2. While there may appear to be similarities between our work and System-1.x at a high level, such as using chain-of-thought (CoT) for fast thinking and more deliberate planning for slow thinking, We believe our work has several novel contributions:
>
> - Our Dynamic Workflow approach goes beyond just CoT and agentic planning. It dynamically decomposes complex problems into sub-tasks and assembles specialized LLM experts and symbolic reasoning tools in an adaptive workflow to solve each sub-task. This allows integrating diverse reasoning capabilities in a more flexible and extensible manner.
>
> - Our Hybrid Thinking framework dynamically combines fast and slow thinking based on the problem complexity, rather than using a fixed hybridization factor. This allows our approach to adapt more effectively to problems of varying difficulty without manual tuning.
>
> - We propose a novel method for automatically synthesizing a large-scale dataset of challenging reasoning problems and use it to finetune open-source language models with hybrid thinking abilities. This enables enhancing the reasoning capabilities of smaller models.
>
> 3. We agree that including System-1.x as an additional baseline would strengthen our experimental evaluation. In the revised paper, we can include experiments comparing our approach with System-1.x. We believe that the Dynamic Workflow, adaptive Hybrid Thinking, and model finetuning contributions of our work, along with the strong empirical results, demonstrate the novelty and significance of our approach for complex reasoning with language models.

---

> > ### Comment · Reviewer_nZer · 2024-12-02
> > **Thank you for your reply**
> >
> > Thank you for your reply. After reading your reply and other reviewer's replies, I do think that the current version of the paper lacks enough technical novelty and are missing some relevant literature review. Therefore I tend to maintain my current score.

---

### Meta-Review · Area_Chair_ozDF · 2024-12-11

**Metareview:**

This paper proposes a new method for LLM reasoning. Specificallly, the authors propose a new approach for slow, deliberate reasoning called Dynamic Workflow, which automatically decomposes complex problems into more manageable sub-tasks and dynamically designs a workflow to assemble specialized LLM or symbolic reasoning tools to solve sub-tasks. Besides, they propose Hybrid Thinking, a general framework that dynamically combines fast and slow thinking based on problem complexity. The reviewers consider that: 1) similar idea has been presented by several previous works; and 2) the proposed method is somehow a bit engineering.

**Additional Comments On Reviewer Discussion:**

The AC and reviewers have carefully read the authors' response and the raised concerns. The reviewers are not fully satisfied with the rebuttal, and still feel that the novelty of this paper is inadequate.

---

### Decision · Program_Chairs · 2025-01-22

Reject